# How Does NIMBYism Influence Residents’ Behavioral Willingness to Dispose of Waste in Centralized Collection Points?—An Empirical Study of Nanjing, China

**DOI:** 10.3390/ijerph192315806

**Published:** 2022-11-28

**Authors:** Qiwen Chen, Hui Liu, Peng Mao, Junjie Qian, Yongtao Tan, Xiaer Xiahou, Peng Cui

**Affiliations:** 1Department of Engineering Management, School of Civil Engineering, Nanjing Forestry University, Nanjing 210037, China; 2School of Engineering, RMIT University, Melbourne, VIC 3001, Australia; 3School of Civil Engineering, Southeast University, Nanjing 211189, China

**Keywords:** centralized collection point, behavioral willingness, NIMBYism, extended theory of planned behavior, structural equation modeling

## Abstract

Residents’ low behavioral willingness to dispose of waste in Centralized Collection Points (CCPs) seriously hinders the operational efficiency in waste collection of CCPs regarded as NIMBY (‘not in my backyard’) facilities. However, fewer researchers notice NIMBY facilities with low hazards. It has been ignored that the NIMBYism may influence behavioral willingness during the operation period persistently. Meanwhile, there is no consistent conclusions on internal factors of waste behavioral willingness, which deserves further study. Therefore, this study took CCPs as a research object and aimed to investigate how NIMBYism influences residents’ behavioral willingness to dispose of waste in CCPs. The extended theory of planned behavior and structural equation modeling approach involving 550 respondents were adopted to conduct the analysis. The results revealed that attitude (β = 0.295, *p* < 0.001), government trust (β = 0.479, *p* < 0.001), and perceived behavioral control (β = 0.222, *p* < 0.001) have statistical positive impacts on behavioral willingness to dispose of waste in CCPs. Perceived risk (β = ‒0.047, *p* = 0.022 < 0.05) can influence behavioral willingness negatively. Additionally, government trust (β = 0.726, *p* < 0.001) exerts a positive impact on attitude. Furthermore, relevant strategies were proposed to enhance residents’ behavioral willingness to dispose of waste in CCPs. This study is expected to inspire the government to formulate policies from the aspects of standards and regulations, resident participation, construction, and publicity. It will provide the government instructive suggestions for the smooth operation of CCPs, and ultimately building a healthy and environment friendly society.

## 1. Introduction

Globally, the phenomenon of waste siege is becoming increasingly severe, which seriously damages the image of cities and potentially threatens the physical and mental health of residents [1,2,3]. In China, the previous strategy of mixed collection, centralized transportation and centralized treatment, commonly known as the three-stage management, can no longer effectively manage the increasing amount of waste in the future [4]. The Centralized Collection Points (CCPs), referring to a collection of color-coded containers with visibly marked signage and sorting information, which specifically collect recyclables, compostable, electronic waste, etc., begin to attract people’s attention. Because CCPs can largely improve the efficiency of waste treatment, facilitate the waste recycling and beautify the urban environment, China had already propagated the slogan “classify waste, and discard waste in CCPs” since the early 1950s, and issued several national and local regulations about proper waste management, which, however, received few satisfactory outcomes. To further improve ecological civilization, the Notice on Comprehensively Carrying out the Sorting of Municipal Solid Waste in Cities at and Above the National Level was issued in 2019. Barrels were replaced with CCPs for waste collection in eight first-tier cities, including Beijing, Shanghai, Guangzhou, Shenzhen, Hangzhou, Nanjing, Xiamen, and Guilin [5].

With the implementation of that policy, key phrases such as “replacing barrels with CCPs” and “waste classification” have become popular in public. However, residents are commonly against this policy by constantly complaining and avoiding classification. Numerous projects relating to this policy failed, or were even ultimately terminated. A presumably important cause of failure is the ignorance of the powerful impact of NIMBYism (Not-In-My-Back-Yard).

NIMBYism refers to a phenomenon that people oppose developmental programs with negative impacts but that serve the public interest in the long run. Specifically, residents may show poor participation and behavioral willingness during the operation of CCPs. For instance, in the Ningbo Zhongxing community, faulty behaviors repeatedly happen, namely, mixed and indiscriminating waste or discarding them in random places than the CCPs. Therefore, if residents can tolerate the inconveniences brought by this policy and even support relating work accordingly, it will greatly promote the implementation of CCPs, and eventually build a healthier and environment friendly society.

However, research on low-hazard NIMBY facilities is inadequate. No matter the site selection or conflict management, prior researchers focus on the early stage, and the management of the later operation stage, especially for NIMBY facilities requiring high willingness during the operation period such as CCPs need further investigation. Researchers ignore the persistent impact of NIMBYism on behavioral willingness during the operation period. Meanwhile, no consistent conclusions on internal factors of waste behavioral willingness have been obtained and deserve to be investigated, too.

Consequently, this research aims to analyze factors influencing residents’ behavioral willingness of disposing of waste in CCPs during the operation period from the angle of internal factors. This study aimed to address two specific questions: (1) Based on the extended theory of planned behavior (TPB), how does NIMBYism influence residents’ behavioral willingness to dispose of waste in CCPs? and (2) What strategies could be proposed to improve residents’ behavioral willingness to dispose of waste in CCPs? To solve those, firstly, this research introduced perceived risk and government trust for extending the TPB and constructed a theoretical model accordingly. Secondly, an empirical study was conducted in three districts in Nanjing. Thirdly, with the structural equation model, this research examined the influencing factors of residents’ behavioral willingness of disposing of waste in CCPs. Finally, corresponding strategies were proposed to improve residents’ behavioral willingness. This study differs from previous ones due to the specificity of research object and its period (operation period). This research is expected to fill the research blank of low-hazard NIMBY facilities (CCPs). In terms of theoretical value, this research provides a basis for conflict management during the operation period of NIMBY facilities and adds more reliable conclusions to the waste behavioral willingness from the perspective of internal factors. In terms of practical value, this study contributes to the smooth operation of CCPs, and ultimately the building of a healthy and environmentally friendly society. The strategies of this research could also provide instructive suggestions for the government to formulate policies concerning the operation of CCPs.

## 2. Literature Review

### 2.1. NIMBYism

With the accelerated development of urbanization and increasing severe environmental problems, NIMBYism has attracted growing attention. As early as the 1970s, researchers proposed the term “NIMBYism”, when they studied mass protests caused by environmental problems. NIMBY facilities are defined by O’Hare as projects with negative impacts but essential to the whole community [6]. NIMBYism refers to the concern that these facilities will bring effects on physical health which triggers people’s resistance [7]. In this study, NIMBYism is defined as residents’ resistance against CCPs.

Existing studies concerning NIMBYism primarily focused on NIMBY facilities with significant hazards such as landfill sites [8,9], nuclear-related facilities [10,11], and solid waste incinerator siting [12,13]. Unfortunately, research on low-hazard NIMBY facilities such as CCPs remains insufficient. The NIMBY facility of the study has its own special characteristics. NIMBYism tends to be expressed in the way of opposing its construction. Previous researchers have been dedicated to this. For instance, Cong [14] explored critical influencing factors for the site selection failure of waste-to-energy projects caused by NIMBYism. Meanwhile, conflict also tends to arise in the pre-construction stage, so a number of studies have focused on conflict management in early stage. Sun [15] studied issues of NIMBYism conflict management from the perspective of stakeholders and provided advanced facility siting response strategies for city managers. Additionally, an early warning system was established to predict and assess the NIMBYism of heavy pollution projects [16]. However, NIMBY facilities such as CCPs or public toilets have their own special characteristics. Not only their construction will be against in the early stages, but also people will express persistent opposition in a way of incorrect use after construction. The management problems of improving NIMBYism during the operation period, then increasing behavioral willingness may still be research deficient. The persistent impact of NIMBYism on behavioral willingness during the operation period is worth investigating further.

### 2.2. Waste Behavioral Willingness

Behavioral willingness greatly reflects one person’s readiness to take actions [17]. In this research, behavioral willingness to dispose of waste in CCPs refers to residents’ readiness of willingly following requirements to classify wastes and discarding them in CCPs.

Numerous studies have shown that external environmental factors, such as information publicity [18], facility conditions [19] and information quality [20], place a significant impact on waste behavioral willingness. In addition to the essential external environmental factors, internal factors, which are less acknowledged by most researchers, are found to equally contribute to it. The existing researches regarding internal factors draw different conclusions. For example, Pivetti [21] and Botetzagias [22] pointed out that moral norms strongly affect people’s attitude, which in turn impacts behavioral willingness. Contrarily, Khan [23] objected the impact of moral norms on behavioral willingness, proposing that subjective norms and consequences awareness are the main factors influencing behavioral willingness. Conclusively, external environmental factors have been sufficiently demonstrated to significantly impact on behavioral willingness. However, because no consistent conclusions on internal factors have been obtained, internal factors require further exploration for their impact.

Most researchers applied the theory of planned behavior (TPB) to investigate behavioral willingness from the angle of internal factors [24,25]. Notably, some scholars have introduced various other factors into TPB based on the study objects. For example, targeting consumers’ food-management behavioral willingness, Soorani and Ahmadvand [26] applied feelings of guilt for extending TPB. Ylä-Mella [27] combined the theory of planned behavior and value-belief-norm for further study of consumers’ behavioral willingness of recycling and re-using mobile phones. According to the actual situation, CCPs are classified as the government project. Meanwhile, residents are the end-users of CCPs. NIMBYism is commonly caused by a lack of government trust [28]. Meanwhile, due to residents’ perceived risk, they appear resistant to CCPs [29]. Hence, these two factors are selected for extending TPB.

To sum up, concerning NIMBYism, fewer attention has been paid to low-hazard NIMBY facilities and study on the management of the later operation stage, especially for NIMBY facilities requiring high willingness during the operation period is inadequate. It has been ignored that NIMBYism could influence behavioral willingness persistently during the operation period. Concerning waste behavioral willingness, internal factors of that also deserve to be investigated further. Therefore, based on the characteristics of CCPs, the study selected perceived risk and government trust to extend TPB. Then, it analyzes factors which pose impacts on residents’ behavioral willingness of disposing of waste in CCPs during the operation period from the angle of internal factors.

## 3. Theory and Hypotheses

### 3.1. Hypotheses Based on TPB

The theory of planned behavior (TPB) was proposed by Ajzen [30], which further explored the relationship between willingness and behavior. Three factors are included in TPB:(1)attitude, namely, positive, or negative personal views on execution;(2)subject norms, namely, a personal perception of social pressure on participation or non-participation;(3)perceived behavioral control, namely, a personal perception of ability to perform specific actions.

So far, TPB has been widely applied in various fields, such as environmental studies [31], health care [32], and marketing [33]. For instance, Zhang [34] found that attitude and perceived behavioral control positively impacted on residents’ willingness to classify wastes. Heidari [35] concluded that attitude and subject norm posed a significant positive impact on waste classification willingness via the TPB. Ma [36] also achieved similar conclusions that attitude, subjective norms, perceived behavioral control are positively related to the willingness to municipal solid waste source-separated collection. Therefore, hypotheses are proposed as the following:

**Hypothesis** **1.**
*Attitude positively affects behavioral willingness to dispose of waste in CCPs.*


**Hypothesis** **2.**
*Subjective norms positively affect behavioral willingness to dispose of waste in CCPs.*


**Hypothesis** **3.**
*Perceived behavioral control positively affects behavioral willingness to dispose of waste in CCPs.*


### 3.2. Hypotheses Based on the Extended TPB

Despite its wide use, TPB has been mostly criticized for focusing on only three factors [37]. Ajzen, one of the founders of the theory, pointed out that new factors could be considered to advance the theory, thereby enhancing the predictive ability of the model [30,38,39], particularly in studying the environment [40]. Since CCPs belong to NIMBY facilities, residents’ NIMBYism needs to be considered during the operation.

The characteristic of NIMBY facilities is public disgust. Perceived risk often appears as a basic influencing factor in the research of the NIMBYism [29]. Meanwhile, NIMBYism is typically caused by a lack of trust in project sponsors (referred to as the government here) [28]. Herein, perceived risk and government trust are considered as two significant factors for extending the TPB to examine the effect of NIMBYism on residents’ behavioral willingness.

#### 3.2.1. Perceived Risk

Perceived risk generally refers to subjective judgments on the possibility of tragic events such as injuries, diseases, and death, which may affect attitudes [41]. Perceived risk has attracted the attention of researchers and policymakers in the extending TPB [42]. Additionally, perceived risk has been widely applied in aspects like purchase willingness [43,44] and NIMBYism [45]. However, few research applies it to analyzing behavioral willingness to dispose of waste in CCPs.

Risks of soil, air and noise pollution exist around CCPs. These risks are both objective and subjective feelings, resulting in public anxiety and resistance [46]. Studies have verified that residents’ perceived risks may negatively affect their attitudes. For example, Choi [47] found in his research that consumers with high perceived risks were inclined to hold a negative attitude toward street food. Arslan [48] verified that perceived risk could negatively influence attitudes on private label consumers in Turkey. Additionally, studies have shown that perceived risks may also negatively affect the behavioral willingness. For example, Li [49] confirmed perceived risks’ negative impact on the willingness of agricultural green production. Liao and Hsieh [50] found that perceived risk is negatively related to consumers’ willingness to purchase Gray-Market Smartphones. Residents who possess higher perceived risks may significantly lower their behavioral willingness to dispose of waste in CCPs. Therefore, hypotheses are proposed as the following:

**Hypothesis** **4.**
*Perceived risk exerts a negative effect on attitude toward behavior to dispose of waste in CCPs.*


**Hypothesis** **5.**
*Perceived risk exerts a negative effect on behavioral willingness of disposing waste in CCPs.*


#### 3.2.2. Government Trust

Citrin and Muste [51] defined government trust as citizens’ confidence that the government will comply with the rules of the game and serve the general interest of the public. In this research, government trust is referred to that residents believe that CCPs built by the government could serve their general interests. Generally, a trustworthy government can assure residents that the NIMBY facilities would bring more benefits than harms, which is especially important for NIMBY facilities like CCPs. When the masses trust the government, they will have a positive attitude and likely obey political leadership without being forced [52], which is also confirmed by Loan [53]. Thus, the study proposed the hypothesis that government trust will positively impact attitude toward the behavior to dispose of waste in CCPs. Moreover, previous researches showed that the people’s behavioral willingness toward hazardous chemicals factory [7], potentially harmful products disposal facilities [54] and other similar NIMBY facilities are positively influenced by government trust. Therefore, hypotheses are proposed as the following:

**Hypothesis** **6.**
*Government trust exerts a positive effect on behavioral willingness to dispose of waste in CCPs.*


**Hypothesis** **7.**
*Government trust exerts a positive effect on attitude toward behavior to dispose of waste in CCPs.*


Perceived risk and government trust are introduced to TPB. Ultimately, the theoretical model for analyzing residents’ behavioral willingness to dispose of waste in CCPs is obtained (Figure 1). The abbreviations used in the study are listed in Table 1.

## 4. Methodology

### 4.1. Questionnaire Design

This research included a questionnaire targeting behavioral willingness to dispose of waste in CCPs. To construct an accurate and effective questionnaire, a small-scale pre-test was conducted in September 2021, which involved 100 residents near CCPs. Additionally, to further ensure the quality of the questionnaire, several management experts were gathered to review the questionnaire, particularly the questions that remained unclear or ambiguous. Ultimately, the final questionnaire was formulated and concrete measurement items are shown in Table 2.

The final questionnaire covers three main sections:(1)the socio-demographics of respondents, including gender, age, education level, personal annual income, residential district;(2)influencing factors of behavioral willingness to dispose of waste in CCPs;(3)behavioral willingness to dispose of waste in CCPs.

All items in the questionnaire were measured on a seven-point Likert scale (1 represents completely disagree and 7 represents completely agree).

### 4.2. Data Collection

Nanjing, the capital city of Jiangsu Province of China, bears a tremendous number of wastes, producing approximately 992.4 million tons in 2020. Although Nanjing was identified as a critical municipal solid waste classification city in 2016 [62], the policies such as “waste classification in the form of bonus points system” achieved unsatisfactory results. Behaviors such as mixing and indiscriminating waste or discarding them in random places than the CCPs are commonly and ubiquitously found in Nanjing. Aiming to analyze the influencing factors of residents’ behavioral willingness to dispose of waste in CCPs, the study selected three districts (Gulou, Jiangning, Jianye) whose number of CCPs ranked the first three [63]. Gulou and Jianye are in the central part of Nanjing, and Jiangning is in the southeast of Nanjing. The large-scale implementation of “replacing barrels with CCPs” in the three districts are since the promulgation of the “*Regulations on the Management of Domestic Waste in Nanjing*”. By the end of May 2021, Jianye, Gulou and Jiangning had built 1040, 1204, and 2316 CCPs respectively, which are the top three in terms of the number of CCPs in Nanjing. Hence, they are selected for study. The sample size of respondents depends on the Cochran formula [64], as showed below.
(1)$$n=\frac{\frac{{p\;(1\;-\;p)\;z}^{2}}{d^{2}}}{1+\frac{1}{N}\;(\frac{{p\;(1\;-\;p)\;z}^{2}}{d^{2}}\;-\;1)}$$
where n denotes the sample size, N represents the total population size of three districts (3,401,900), z is the standard value for confidence level (1.96), d is the allowable error value (0.05), and p is the estimated proportion of an attribute present in the population (0.5). The Cochran formula result shows that at least 384 samples are required to conduct the survey.

The survey was conducted in the fourth quarter of 2021. The questionnaires were distributed in the form of one-to-one to ensure that each participant fully understands the questionnaire. Ultimately, 550 questionnaires were collected, and all questionnaires are valid, leading to an effective rate of 100%. Table 3 shows that the number of questionnaires distributed in the three districts is roughly the same. Of the 550 respondents, 318 are female, and 232 are male. Over three quarters are in the age group between 18 and 34. Approximately 77.3% respondents have received higher education from universities. The respondents are widely distributed in different income and age groups can fairly reflect the behavioral willingness of more kinds of residents. Most respondents are generally well-educated, indicating that they have a higher possibility to fully understand the questionnaire.

### 4.3. Data Analysis

The structural equation model, an objective mathematical model, combines complex path models with latent factors [65]. It can construct models under complex conditions involving hidden variables, independent and multiple dependent variables, as well as variable errors. It is suitable to analyze the relationship between complex factors based on large samples, especially on satisfaction [66,67] and willingness [68,69]. It has been widely applied in numerous fields such as sustainable infrastructure [70] and waste separation [53]. Therefore, the structural equation model is primarily chosen for studying influencing factors of residents’ behavioral willingness to dispose of waste in CCPs. Firstly, SPSS 24.0 was employed to conduct descriptive statistics analysis and reliability analysis. Secondly, Amos 24.0 was adopted to conduct confirmatory factor analysis to test the validity of the data. Finally, the structural model was constructed and modified, achieving *p*-value to verify the proposed hypothesis.

## 5. Results

### 5.1. Descriptive Statistics

The mean values (M), standard deviations (SD), skewness value (SK), and kurtosis value (K) of construct items are listed in Table 4. Since the data of the structural equation model generally fit normal distribution, this research conducted the normality assumption test. Severe, abnormally distributed samples whose absolute SK value is greater than three, and K value is over ten are not accepted [71]. Table 4 reveals that the maximum value of SK is −0.310, and the minimum value of that is −1.320. The K value ranges from −0.199 to 2.033. No SK value and K value falls outside the constraint scope. Further research could be conducted.

### 5.2. Reliability Analysis

Cronbach’s α was employed to test the internal consistency of the constructs. If Cronbach’s α is greater than 0.7, reliability is acceptable [72]. With the assistance of SPSS 24.0, overall Cronbach’s α and each construct’s α values were obtained. Table 5 shows that the Cronbach’s α of attitude, subjective norm, perceived behavioral control, perceived risk, government trust, and behavioral willingness are 0.949, 0.907, 0.915, 0.906, 0.965, and 0.932 respectively. Overall, Cronbach’s α is 0.953. All α values exceeded 0.7, indicating a considerately high reliability of the collected data. Additionally, the KMO value was 0.952, and the *p*-value was below 0.001, indicating that the data was suitable for factor analysis [18].

### 5.3. Validity Analysis

Amos 24 is applied to conduct confirmatory factor analysis to test the validity of the collected data. Table 6 shows that, absolute fit, incremental fit, and parsimonious fit of measurement model all met the acceptable thresholds of model fitness [73]. Additionally, items with standardized factor loadings below 0.5 should be eliminated [74]. Table 7 demonstrates that all standardized factor loadings greatly exceed 0.5. Thus, no item is eliminated. The initial measurement model is shown in Figure 2. Besides the factor loading test, converged and discriminant validity tests were additionally conducted. All the average variances extracted passed the cut-off value of 0.5 [75]. All the combined reliability passed the cut-off value of 0.7 [76]. Thus, the collection data showed good convergence validity.

Subsequently, a discriminant validity test was conducted. It shows that the square root of each average variances extracted exceeds corresponding inter-construct correlations in related the rows and columns, indicating adequate discriminant validity for all constructs [77]. Table 8 demonstrates that the data is equipped with good discriminative validity. Collectively, the data used in the structural equation model has passed the reliability and the validity test.

### 5.4. Results of Model Modification and Test

On the basis that the initial measurement model has passed the reliability and the validity tests, the structural model (model 1) is constructed accordingly. As shown in Table 6, X^2^/df value is 3.075, which is acceptable but not highly satisfactory [78]. Therefore, model 1 remained to be modified. It is pointed out that the relationship between measurement variables and potential variables, or among measurement variables may affect the model fitting. Especially, those parameters that are not estimated may be statistically significant. Parameters with high MI values can be freed from constraints, which indicates improved correlations between variables [79]. Considering its practical significance, a new correlation was constructed between the two error items with the highest MI value (e1 <=> e2, MI = 22.327), obtaining model 2. Specific results of model 2 are demonstrated in Table 9.

Fitness of model 2 lies within a satisfying range after modification. Therefore, model 2 was determined as the final model. The standardized estimation coefficient of each path is shown in Figure 3, which indicates the contribution weight of each influencing factor toward the behavioral willingness to dispose of waste in CCPs.

Specific hypothesis results are shown in Table 10. It shows that attitude (β = 0.295, *p* < 0.001) and perceived behavioral control (β = 0.222, *p* < 0.001) have statistical positive impacts on behavioral willingness to dispose of waste in CCPs, supporting H1 and H3. Subjective norms (*p* = 0.07 > 0.05) can’t influence behavioral willingness statistically, thus rejecting H2. Meanwhile, perceived risk can exert a positive impact on attitude (β = 0.072, *p* = 0.028 < 0.05) toward behavior to dispose of waste in CCPs and a negative impact on behavioral willingness to dispose of waste in CCPs (β = −0.047, *p* = 0.022 < 0.05). Hence, H5 was supported while H4 was rejected. Residents’ attitude (β = 0.726, *p* < 0.001) and behavioral willingness (β = 0.479, *p* < 0.001) are also positively affected by government trust, supporting H6 and H7.

## 6. Discussion

### 6.1. Attitude

Attitude has been confirmed to positively affect behavioral willingness in this research. Specifically, the more positive subjective perspectives residents hold, the higher behavioral willingness they prefer [80,81]. The conclusion that attitude can be used to predict specific behavioral willingness was also obtained by Campbell [82] and Sherman and Fazio [83] as early 20th century. Additionally, though it has been manifested that government trust places a statistical impact on the attitude of residents toward waste classification [53]. This research further extends the above result from the aspect of perceived risk, demonstrating that perceived risk also plays an important role.

### 6.2. Subjective Norm

Subjective norm has been surprisingly found to place an insignificant impact on residents’ behavioral willingness to dispose of waste in CCPs. Contrary to TPB proposed by Ajzen [30], that families, friends and residents in communities voluntarily dispose of waste in CCPs could not notably improve respondents’ behavioral willingness. This finding is in consistent with the study of Coşkun and Özbük [84]. It could be presumably explained by that subjective norm reflects social pressure, which is challenging to be quantitatively measured. Additionally, many studies have shown that the relationship between subjective norm and behavioral willingness tends to be weaker than other factors in TPB [85,86]. According to the practical situation, regulations on NIMBY facilities such as CCPs in Nanjing have not been constructed sufficiently and strictly. Respondents assumed that low behavioral willingness to dispose of waste in CCPs is of little significance to the society and should not be criticized and corrected by their friends and other. Ma [36] additionally pointed out the problem.

### 6.3. Perceived Behavioral Control

Consistent with the findings of Knussen [87], perceived behavioral control positively affects residents’ behavioral willingness to dispose of waste in CCPs. As a new variable added into the theory of reasoned action, perceived behavioral control is utilized to reflect perceptions of both internal (knowledge, skills, willpower, etc.) and external (time, convenience, cooperation, etc.) factors. It has been exhibited that perceived behavioral control exerts positive impacts on behavioral willingness [88,89]. In terms of internal factors, since the policies of CCPs are newly implemented, it is reasonable that residents are not instantly equipped with right awareness about waste disposal and high behavioral willingness. In terms of external factors, access to CCPs is restricted, particularly in urban areas. For instance, only approximately 1.48 CCPs are found per average of 1 square kilometer in the Jiangning district of Nanjing. A low density of CCPs severely hinders the improvement of perceived behavioral control.

### 6.4. Perceived Risk

An interesting finding in this study is that perceived risk places a negative effect on behavioral willingness, but a positive effect on attitudes. Perceived risk can negatively influence behavioral willingness, which is consistent with most of studies. High-risk perceived residents are reluctant to approach, not to mention dispose of waste in CCPs because they worry that CCPs may pose an unpleasant odor or health threat to their lives, etc. Meanwhile according to hypothesis results (Table 10), perceived risk exerts the lowest impact on behavioral willingness compared with other factors. However, according to the research of Lee [90] toward the online banks and Hamid and Bano’s [91] research on how COVID-19 influences tourism, perceived risk places a crucially important role on behavioral willingness. This inconsistency can be possibly explained by that the olfactory and visual unpleasantness brought by CCPs are more tolerated than monetary losses and viruses, which exert more influencing power to residents. Economic recession and the threat of viruses will pose higher perceived risks.

This study notices that perceived risk exerts a surprisingly positive effect on attitudes, although it is generally accepted that perceived risk has a negative effect on attitudes from the definition of perceived risk [48,92]. Still, some scholars have reached consistent conclusions with this study that perceived risk can have a positive effect on attitudes. As Gstaettner [93] has found in their study of nature travel, the tourists’ perceived risks posed by beaches would positively influence attitudes. According to them, perceived risk is essential to travel, and a certain level of it can increase amusement in their experience. Palau-Saumell [94] also pointed out that in the background of the COVID-19 epidemic, the perceived risk toward infection and illness will lead to the improvement of attitudes towards local foods when individuals are concerned about food contamination during transportation. The adventurous nature of travel and the safety of local foods contribute to the positive effect of perceived risk on attitudes. In this study, CCPs are a class of NIMBY facilities with great environmental benefits. Numerous studies have confirmed that an increase in the level of individual risk perception of environmental problems will remarkably raise the level of individual concern about environmental problems [95]. Therefore, residents with higher perceived risk of CCPs will, to a certain extent, be more concerned about the quality of their living and community environment, and accordingly show a positive trend in their attitudes toward CCPs, which are considered highly environmentally beneficial.

### 6.5. Government Trust

Government trust could positively influence residents’ behavioral willingness to dispose of waste in CCPs and its attitude, which is consistent with Loan [53]. If residents trust that NIMBY facilities built by the government such as CCPs would serve their general interests, residents’ behavioral willingness will increase accordingly. Many researches based on TPB demonstrate that attitude is of prime importance on behavioral willingness [96]. Contrarily, in this extended TPB, government trust poses a more powerful influence on behavioral willingness than attitude. Additionally, our study shows that government trust strongly exerts positive impacts on attitude. Regarding NIMBY facilities such as CCPs, if the government can assure the public that they are able to address the problems brought by CCPs, showing them the resolves and efforts, residents’ government trust will grow significantly and their behavioral willingness will be significantly improved accordingly.

### 6.6. Strategies for Promoting Residents’ Behavioral Willingness

From the results of the structural equation model (shown in Figure 3 and Table 10), it can be found that perceived behavioral control, government trust, attitude, and perceived risk can be considered as factors which would statistically affect residents’ behavioral willingness. Herein, strategies relating to these factors for promoting residents’ behavioral willingness are established [97] (as shown in Figure 4), which are divided into 4 modules (module1 standards and regulations, module 2 resident participation, module 3 construction of CCPs, module4 publicity of CCPs). In this system, as users of CCPs, residents will participate in the construction and publicity of CCPs by giving feedbacks, while standards and regulations will guide the other three modules. Additionally, the module3 contains module 3-1 and module 3-2. Similarly, module 4 includes module 4-1 and module 4-2.

Due to powerful impact of government trust on residents’ behavioral willingness, this study proposed strategies from the aspects of module1 (standards and regulations) and module2 (resident participation) targeting government trust. The main power of the Chinese government trust system is emanated from standards and regulations [98,99]. Specifically for NIMBY facilities such as CCPs, it is recommended that the government improve the regulations for the construction of CCPs and the supervision and punishment mechanisms closely related to the interests of all parties. The lack of uniform standards for waste classification could easily confuse new residents about waste classification, which is not conducive to the improvement of their willingness to dispose of waste in CCPs. Therefore, establishing uniform standards for waste classification as soon as possible is needed. There have been extensive studies proving that government trust is closely linked to public participation [100]. To this regard, it is suggested that the government, jointly with the community, collect and adopt residents’ appropriate opinions for decision-making by holding hearings, committees, and establishing opinion feedback platforms. Concerning the CCPs feedback platform, it is recommended to assign a commissioner to collect platform views, and respond to each opinion adopted or rejected, with corresponding measures after adoption and reasons for refusal.

Perceived behavioral control has statistical positive impacts on behavioral willingness to dispose of waste in CCPs. Hence, these proposed strategies from the aspects of module 3-1 accessibility of CCPs and module 4-1 publicity of waste classification knowledge for the improvement of perceived behavioral control. Perceived behavioral control reflects various aspects of internal (knowledge, skills, willpower, etc.) and external (time, convenience, cooperation, etc.) factors. From an external perspective, CCPs and their supporting construction can be improved to enhance the accessibility of CCPs and make them more convenient for residents to dispose of waste. The following strategies can be adopted specifically: (1) increasing the density of CCPs and (2) setting CCPs distribution maps. From the internal perspective, the study can promote the publicity of waste classification knowledge to improve the residents’ own internal capacity of waste classification. It was also found that residents showed a lower mean score for the question “I know how to classify wastes as expected and dispose of waste accordingly in CCPs” than for the other questions on perceived behavioral control (MPBC3 = 4.98). Residents may have blind spots in waste classification knowledge, so it is recommended to strengthen the publicity of concerning knowledge, specifically by adopting strategies such as painting colorful wall relating to waste classification and exoteric knowledge (blue bins for money; green bins for perishable, red bins for toxic and harmful waste, gray bins for no-one-would-want). It is also suggested to include waste classification in the general education curriculum.

Perceived risk influences behavioral willingness to dispose of waste in CCPs statistically. Hence, this proposed strategies from the aspects of module 3-2 sanitation of CCPs, module 4-1 publicity of waste classification knowledge and module 4-2 clarify the negative impact of CCPs to improve perceived risk. Because perceived risk is closely linked to familiarity and media exposure [101,102]. Therefore, the publicity of waste classification knowledge, which is a strategy of perceived behavioral control, can also improve perceived risk. Meanwhile, it was noticed that the mean values of PR2, PR3 and PR4 were significantly lower than the combined questionnaire items (MPR2 = 4.89, MPR3 = 4.71, MPR4 = 4.62). This indicates that there are still many false negative perceptions of CCPs among residents and further popularization of science is still needed to clarify the negative effects of CCPs. Therefore, it is suggested that experts in the relevant fields can provide scientific explanations on the negative effects of CCPs and remove their perception biases toward CCPs. It is encouraged for bloggers to share their own experiences in disposing of waste in CCPs through mainstream media (Weibo, Jitterbug, B-site) to positively guide the direction of public opinion and build residents’ correct perception of NIMBY facilities like CCPs. In addition, the following strategies are proposed concerning the sanitation of CCPs to address the risks of unpleasant odors and risks to the community environment and residents’ health associated with CCPs: (1) routinely cleaning of CCPs and garbage collection trucks to reduce unpleasant odors; (2) assembling facilities such as insect-repelling lamps and draft fans to reduce the negative environmental impact of CCPs on the community; (3) installing facilities such as hand sinks and disinfectant near CCPs to reduce the spread of bacteria in waste and protect the residents’ health.

Because government trust and perceived risk can not only directly influence behavioral willingness, but also indirectly influence behavioral willingness positively by influencing the attitude of them, all of the aforementioned strategies on perceived risk and government trust can noticeably improve residents’ attitudes. In summary, this study develops strategies to effectively improve residents’ willingness to dispose of waste in CCPs, which would eventually advance our living conditions ecologically.

## 7. Conclusions

It is imperative to implement the policy of “replacing barrels with CCPs” for a greener environment of human beings, which however, unfortunately provokes residents’ resistance. The smooth operation of CCPs relies on a higher behavioral willingness of residents to dispose of waste in CCPs properly. This research aims to analyze the influencing factors of residents’ behavioral willingness to do so and provide strategies for improvement. This research constructs a theoretical model by introducing perceived risk and government trust into TPB. Structural equation modeling was then applied to examine the impacts of influencing factors. The results showed that: (1) Perceived risk and government trust affect behavioral willingness directly. In addition, they could positively impact attitude toward the behavior of disposing of waste in CCPs, then indirectly influencing behavioral willingness through attitude. (2) Perceived behavioral control positively influences residents’ behavioral willingness to dispose of waste in CCPs. (3) Behavioral willingness is not statistically affected by subjective norms. Furthermore, targeted strategies are proposed for improving government trust, perceived behavioral control, perceived risk, and attitude, ultimately achieving the smooth operation of CCPs. Specific strategies include four modules (standards and regulations, resident participation, construction of CCPs, and publicity of CCPs).

This research provides both theoretical and practical implications. From the perspective of theory, this research complements the insufficient literature of NIMBY facilities’ conflict management during the operation period, especially NIMBY facilities with low hazards. What’s more, it is expected to provide more reliable conclusions for the waste behavioral willingness from the angle of internal factors. From the perspective of practice, our study will be instructive in improving residents’ behavioral willingness to dispose of waste in CCPs and providing a reference for the government to guide residents properly. It also promotes the implementation of the polices of “replacing barrels with CCPs” and “waste classification”, inducing a healthy, and environmentally friendly society. Additionally, our finding that subject norm shows no statistical impact on behavioral willingness in this research, could provoke further academic discussions about the relation between them. The interesting findings about perceived risk in our research may give more interesting inspiration to peers. Although this is an empirical study of Nanjing, the research findings could be applied in other regions of China.

Though the study has gained interesting and promising findings, there are still some unavoidable limitations. The investigation in this research was only conducted in Nanjing. More representative samples from other provinces should be included in the future to obtain more reliable results. At the same time, suggestions proposed in this study could be validated through introducing other methods.

## Figures and Tables

**Figure 1 ijerph-19-15806-f001:**
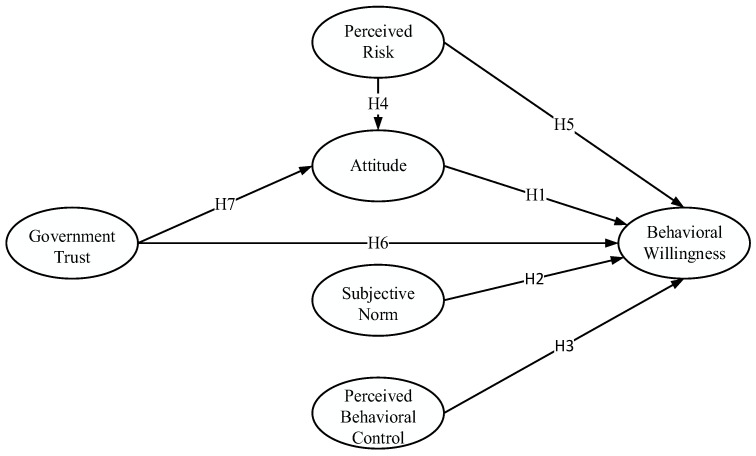
Theoretical model of the extended theory of planned behavior.

**Figure 2 ijerph-19-15806-f002:**
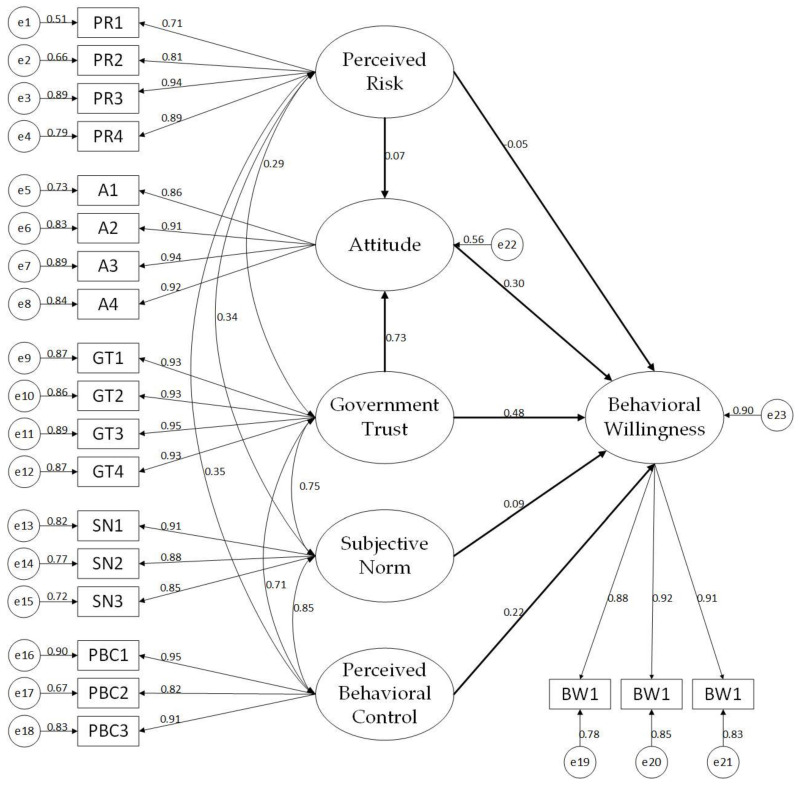
Initial measurement model of residents’ behavioral willingness.

**Figure 3 ijerph-19-15806-f003:**
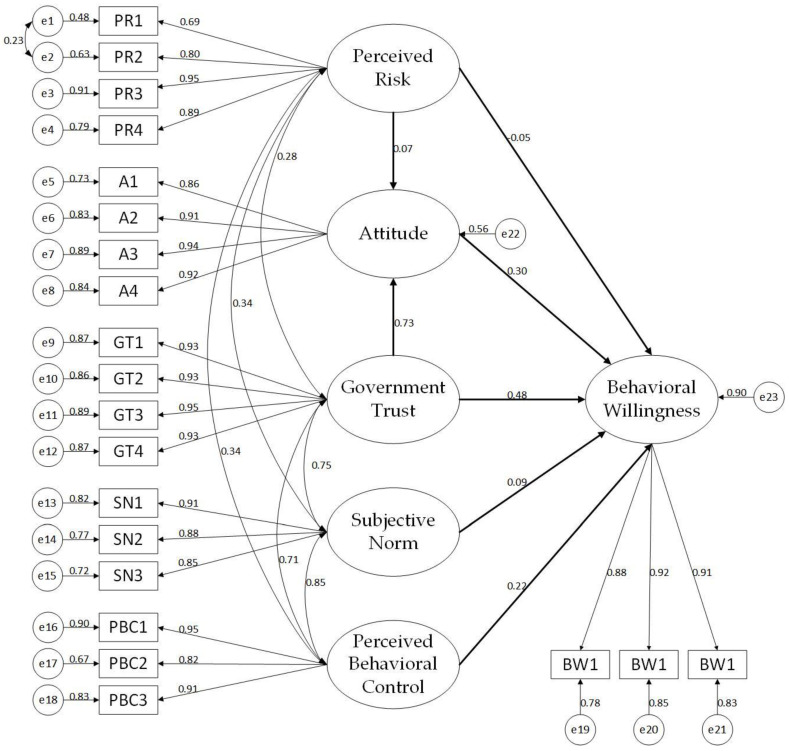
Model 2 of residents’ behavioral willingness.

**Figure 4 ijerph-19-15806-f004:**
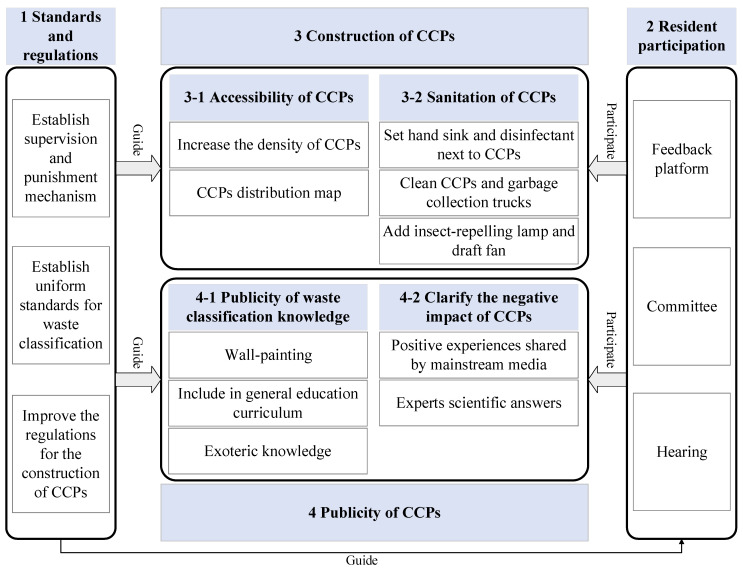
Strategies for promoting residents’ behavioral willingness.

**Table 1 ijerph-19-15806-t001:** List of abbreviations.

Abbreviation	Full Name
CCPs	Centralized Collection Points
A	Attitude
SN	Subjective Norm
PBC	Perceived Behavioral Control
PR	Perceived Risk
GT	Government Trust
BW	Behavioral Willingness

**Table 2 ijerph-19-15806-t002:** Measurement items of the constructs.

Latent Factors	Code	Measurement Items	References
A	A1	That disposing of waste in CCPs would benefit the environment.	[30,55,56]
	A2	I hold a positive attitude toward the behavior of disposing of waste in CCPs.	
	A3	I support the policy on disposing waste in CCPs.	
	A4	That disposing of waste in CCPs should be promoted nationwide.	
SN	SN1	My families and friends always dispose of waste in CCPs.	[30,55]
	SN2	Residents in my community always dispose of waste in CCPs.	
	SN3	My families and friends support the policy on waste disposal in CCPs.	
PBC	PBC1	CCPs are located within the walking distance from my house.	[30,55,57]
	PBC2	That disposing of waste in CCPs is not difficult for me to follow.	
	PBC3	I know how to classify wastes as expected and dispose of waste accordingly in CCPs.	
PR	PR1	I’m concerned that the CCPs would be malodorous and develop poor-quality air.	[58,59]
	PR2	I think the prices of houses adjacent to CCPs will fall.	
	PR3	I am concerned that CCPs will bring health problems in the long run to nearby residents.	
	PR4	I think CCPs may be potentially harmful to the community.	
GT	GT1	I believe that CCPs built by the government would bring more benefits than disadvantages.	[60,61]
	GT2	I trust the government for supervising the operation of CCPs, and solving the existing problems.	
	GT3	Residents’ willingness would effectively influence the government’s decision-making.	
	GT4	I trust that the government, with effective legal instruments, has been making efforts in solving problems brought by CCPs.	
BW	BW1	I am willing to dispose of waste in CCPs as required.	[19,30,56]
	BW2	I plan to dispose of waste in CCPs in the future.	
	BW3	If I have the chance, I will encourage someone important for me to dispose of waste in CCPs.	

**Table 3 ijerph-19-15806-t003:** Demographic characteristics of the participants.

Variable	Classification	NO.	Percentage
Gender	Male	232	42.2%
	Female	318	57.8%
Age	<18	19	3.5%
	18–34	445	80.9%
	35–59	77	14.0%
	≥60	9	1.6%
Education level	Primary school or below	4	0.7%
	Junior high school	16	2.9%
	High school (secondary technical school)	36	6.5%
	College for professional training	69	12.5%
	University and above	425	77.3%
Personal annual income	≤3000	317	57.6%
	3001–6000	103	18.7%
	6001–10,000	65	11.8%
	10,001–15,000	31	5.6%
	>15,000	34	6.2%
Belonging district	Gulou	175	31.8%
	Jiangning	196	35.6%
	Jianye	179	32.5%

**Table 4 ijerph-19-15806-t004:** Descriptive statistics of construct items.

Construct Item	Overall Sample (*n* = 550)			
	M	SD	SK	K
A1	6.08	1.128	−1.320	2.033
A2	5.97	1.123	−1.094	1.480
A3	6.00	1.124	−1.150	1.559
A4	5.96	1.162	−1.104	1.200
SN1	5.21	1.445	−0.410	−0.478
SN2	5.21	1.441	−0.421	−0.396
SN3	5.50	1.300	−0.541	−0.199
PBC1	5.21	1.380	−0.411	−0.269
PBC2	5.40	1.296	−0.439	−0.289
PBC3	4.98	1.486	−0.310	−0.488
PR1	5.05	1.583	−0.549	−0.230
PR2	4.89	1.538	−0.349	−0.331
PR3	4.71	1.633	−0.324	−0.436
PR4	4.62	1.696	−0.337	−0.490
GT1	5.63	1.233	−0.748	0.500
GT2	5.59	1.229	−0.623	0.245
GT3	5.71	1.197	−0.847	0.979
GT4	5.68	1.213	−0.806	0.784
BW1	5.73	1.134	−0.786	0.900
BW2	5.66	1.189	−0.693	0.420
BW3	5.61	1.197	−0.620	0.304

Note: M = mean; SD = standard deviation; SK = skewness value; K = kurtosis value.

**Table 5 ijerph-19-15806-t005:** Results of the reliability test.

Variable	No. of Items	Cronbach’s α
A	4	0.949
SN	3	0.907
PBC	3	0.915
PR	4	0.906
GT	4	0.965
BW	3	0.932

Note: a Overall Cronbach’s α = 0.953; KMO = 0.952; *p* < 0.001.

**Table 6 ijerph-19-15806-t006:** Goodness-of-fit of the initial measurement model.

	Absolute Fit				Incremental Fit				Parsimonious Fit	
Goodness-of-Fit	X^2^/df	GFI	AGFI	RMSEA	TLI	IFI	CFI	NFI	PNFI	PGFI
Initial measurement model	3.075	0.912	0.885	0.061	0.966	0.972	0.972	0.959	0.804	0.695
Level of acceptance fit	1–5 accepted	>0.90	>0.80	<0.08	>0.95	>0.90	>0.90	>0.90	>0.50	>0.50

Note: X^2^/df = chi-square divided by the degrees of freedom; GFI = goodness of fit index; AGFI = adjusted goodness of fit index; RMSEA = root mean square error of approximation; TLI = Tucker–Lewis index; IFI = incremental fit index; CFI = comparative fit index; NFI = normed-fit index; PNFI = parsimonious normed fit index; PGFI = parsimonious goodness-fit-index.

**Table 7 ijerph-19-15806-t007:** Converged validity and standardized factor loadings of the initial measurement model.

Construct	Item	FL	CR	AVE
A	A1	0.856 ***	0.973	0.899
	A2	0.913 ***		
	A3	0.945 ***		
	A4	0.916 ***		
SN	SN1	0.906 ***	0.908	0.766
	SN2	0.875 ***		
	SN3	0.848 ***		
PBC	PBC1	0.948 ***	0.921	0.796
	PBC2	0.816 ***		
	PBC3	0.909 ***		
PR	PR1	0.714 ***	0.903	0.723
	PR2	0.810 ***		
	PR3	0.943 ***		
	PR4	0.891 ***		
GT	GT1	0.935 ***	0.965	0.872
	GT2	0.925 ***		
	GT3	0.946 ***		
	GT4	0.931 ***		
BW	BW1	0.883 ***	0.933	0.822
	BW2	0.924 ***		
	BW3	0.913 ***		

Note: *** *p* < 0.001; FL = factor loading; CR = combined reliability; AVE = average variances extracted.

**Table 8 ijerph-19-15806-t008:** Correlation matrix and discriminant validity for the constructs.

	A	SN	PBC	PR	GT	BW
A	(0.948)					
SN	0.595 **	(0.875)				
PBC	0.553 **	0.792 **	(0.892)			
PR	0.278 **	0.317 **	0.337 **	(0.850)		
GT	0.703 **	0.705 **	0.689 **	0.282 **	(0.934)	
BW	0.770 **	0.744 **	0.752 **	0.278 **	0.855 **	(0.907)

Note: ** *p* < 0.01.

**Table 9 ijerph-19-15806-t009:** Goodness-of-fit of model 2.

	Absolute Fit				Incremental Fit				Parsimonious Fit	
Goodness-of-Fit	X^2^/df	GFI	AGFI	RMSEA	TLI	IFI	CFI	NFI	PNFI	PGFI
Model 2	2.953	0.916	0.889	0.060	0.968	0.974	0.974	0.961	0.801	0.694
Level of acceptance fit	1–3 perfect; 1–5 accepted	>0.90	>0.80	<0.08	>0.95	>0.90	>0.90	>0.90	>0.50	>0.50

**Table 10 ijerph-19-15806-t010:** Hypothesis results.

Hypothesized Paths	β-Value	*p*-Value	Test Results
H1: A→BW	0.295	0.000	Supported
H2: SN→BW	0.086	0.070	Reject
H3: PBC→BW	0.222	0.000	Supported
H4: PR→A	0.072	0.028	Reject
H5: PR→BW	−0.047	0.022	Supported
H6: GT→BW	0.479	0.000	Supported
H7: GT→A	0.726	0.000	Supported

## Data Availability

The data presented in this study are available on request from the corresponding author. The data are not publicly available due to privacy.

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
