# Peer review of "How Does NIMBYism Influence Residents’ Behavioral Willingness to Dispose of Waste in Centralized Collection Points?—An Empirical Study of Nanjing, China"

_ijerph, 2022, doi:10.3390/ijerph192315806_

Round 1

Reviewer 1 Report

“How Does NIMBYism Influence Residents’ Behavioral Willingness to Dispose of Waste in Centralized Collection Points? — An Empirical Study of Nanjing, China” submitted to International Journal of Environmental Research and Public Health aim to improve the residents’ behavioral willingness to dispose. The topic has practice value for sustainable development. The paper also has a complete structure. However, there are still some problems existed in this study. The detailed comments are presented below.
1.     Abstract is not well written. A good abstract should include research background, novelty and contribution, research methods, results, and implications. More quantitative results and policy implications should be added in the Abstract. 
2.     Introduction: The novelty and practical value are not clearly mentioned. The knowledge gap needs to be clearly addressed in introduction.
3.     It is difficult to summarize your research contributions according to literature review section. Please add more details about the previous works.
4.     Results analysis writing could be much more concise and with clear logic. For example, in section5.1, I suggest authors to add quantitative results and analysis.
5.     It is highly recommended to improve your figures quality and level. Most of your figures are displayed unprofessionally.
6.     In the conclusion part, given policy recommendations are also too generic.

Thus, based on these feedbacks, I suggest a major revision on this manuscript. 

Reviewer 2 Report

<minor revision>

Paying attention to the operation period of NIMBY (‘not in my backyard’) facilities is important to build the green environment. This paper investigates how NIMBYism influences residents’ behavioral willingness to dispose of waste in CCPs (Centralized Collection Points). The hypotheses proposed are well explained, as well as the consequently, results, discussion and conclusion sections are described in an appropriate way. However, some problems should be solved or explained before publication.

1) Page 8 4.2 Data collection, general description and the reason why those districts are chosen is suggested to illustrate. 

2) Figure 4 is confused, what is the meaning of the different colors? Please indicate.

3) In the Literature Review section, authors have mentioned that existing studies concerning NIMBYism primarily focused on facility site selection and conflict management. However, from my perspective, the study on residents’ behavioral willingness to dispose of waste in CCPs belongs to the scope of conflict management.  Since conflict management has been focus on, it won't provide any useful information for the reader to study.

4) Given that there many abbreviations in the paper, I suggest to add a list of abbreviations.

5) Some language issues should be modified, such as try to avoid “I” and “we” in the manuscript; Some words should follow the statistical standards, for example, “statistically significant” would be more appropriate instead of using “significant impact” and so on.
